# Metastatic Penile Squamous Cell Carcinoma Responsive to Enfortumab Vedotin

**DOI:** 10.3390/ijms242216109

**Published:** 2023-11-09

**Authors:** Catherine C. Fahey, Caroline A. Nebhan, Sally York, Nancy B. Davis, Paula J. Hurley, Jennifer B. Gordetsky, Kerry R. Schaffer

**Affiliations:** 1Department of Medicine, Vanderbilt University Medical Center, Nashville, TN 37232, USA; catherine.fahey@vumc.org (C.C.F.);; 2Tennessee Valley Healthcare System, Veterans’ Affairs, Nashville, TN 37232, USA; 3Intermountain Health, Murry, UT 84107, USA; 4Vanderbilt-Ingram Cancer Center, Nashville, TN 37232, USA; 5Department of Urology, Vanderbilt University Medical Center, Nashville, TN 37232, USA; 6Department of Pathology, Microbiology and Immunology, Vanderbilt University Medical Center, Nashville, TN 37232, USA

**Keywords:** penile squamous cell carcinoma, Nectin-4, antibody-drug conjugate, targeted therapy, metastatic penile cancer

## Abstract

Penile squamous cell carcinoma is a rare disease with very limited data to guide treatment decisions. In particular, there is minimal evidence for effective therapies in the metastatic setting. Here, we present a case of metastatic penile squamous cell carcinoma with response to the Nectin-4 inhibitor enfortumab-vedotin-ejfv (EV). EV was selected due to the evidence of the high expression of Nectin-4 in squamous cell carcinomas, including penile carcinoma. The patient had both radiographic and symptomatic improvement after two cycles of treatment, despite having been treated with multiple prior lines of traditional chemotherapy. This case provides support for the use of antibody–drug conjugates (ADC), including EV, in this disease with few other options in the advanced setting. Further studies examining Nectin-4 and ADCs in penile squamous cell carcinoma should be completed, as high-quality evidence is needed to guide treatment after initial progression for these patients.

## 1. Introduction

Penile squamous cell carcinoma (SCC) is a relatively rare malignancy in the United States, with an expected incidence of 2000 cases and 500 deaths in 2022 [1]. Cases are more common in African and South American countries, where penile carcinoma can represent up to 10% of malignancies in men [2]. A paucity of cases renders therapeutic development in randomized clinical trials for this cancer difficult, and the five-year overall survival for metastatic disease approaches 0% [3,4]. Reflecting this challenge, the current recommended penile cancer-specific therapies in the National Comprehensive Cancer Network (NCCN) guidelines are based on data published 10+ years ago [5]. There are several ongoing clinical trials exploring novel treatment approaches for this rare cancer, including trials investigating single agents and combination therapies of immunotherapy, PARP inhibitors, traditional chemotherapies, VEGF inhibitors, EGFR inhibitors, and engineered T cells [6]. Currently, however, compelling data for recurrent/refractory disease is lacking, and there is a need for additional effective systemic therapies for advanced penile cancer. 

Penile SCC can be HPV-mediated or non-HPV mediated, with upwards of 40% of cases showing p16 positivity [7]. Additional risk factors for the development of penile SCC include phimosis, lack of circumcision, tobacco use, poor hygiene, genital warts, urinary tract infections, and obesity [8]. Molecular changes commonly seen in penile carcinogenesis include alterations in RAS and MYC, changes to E-cadherin expression, and changes in the p53 and Rb pathways [2]. Additionally, Nectin-4 has been shown to be highly expressed in squamous cell carcinoma, including penile SCC [9]. Nectin-4 has multiple cellular roles, including as a member of the adherens junction [10] and as a ligand for immune checkpoint inhibition [11]. In a cohort of 57 cases of penile SCC, 89% had >25% of cells show positive staining for Nectin-4, versus no staining in nonmalignant penile tissue. Given this biologic finding, there is a logical rationale for the therapeutic potential of Nectin-4 targeted agents in this rare patient population.

First granted accelerated FDA approval in December 2019, the Nectin-4-directed antibody–drug conjugate (ADC) Padcev™ (enfortumab vedotin-ejfv) (EV) is approved for use in advanced or metastatic urothelial cancer after progression with a PD-1 inhibitor and platinum-based chemotherapy (if eligible) [12]. The ADC binds to Nectin-4 on the cell surface and is internalized, allowing for the proteolytic cleavage of the microtubule-disrupting agent monomethyl auristatin E (MMAE), leading to cell death. Here, we present a case of metastatic penile squamous cell carcinoma with response to EV and discuss the biologic rationale supporting a role for this therapeutic class in this rare cancer.

## 2. Case Description

A 67-year-old man presented with penile pain and bleeding after a trauma. His medical co-morbidities were notable for hypertension and prior tobacco use. The exam showed a painless, fluctuant nodule, <1 cm in diameter, located at the corona of the glans penis. A percutaneous biopsy of the penile lesion revealed p16-positive, HPV-associated, basaloid squamous cell carcinoma. A pelvic MRI revealed an indeterminate nodular density at the distal penile head and a CT urogram revealed pulmonary lesions concerning for metastatic disease. The patient underwent a partial penectomy, which showed pT3 disease with tumor invasion into the lamina propria, corpus spongiosum, corpus cavernosum, and tunica albuginea. Lymphovascular invasion was identified. The surgical margins were positive; the tumor extended to the skin resection margin. PET-CT imaging revealed multiple bilateral pulmonary nodules with strong F-fluorodeoxyglucose (FDG) avidity, the largest measuring 1.1 cm. Hypermetabolic activity was also observed along the residual distal penile shaft. A bronchoscopy with biopsy confirmed metastatic p16-positive SCC.

After metastatic disease was confirmed, the patient was treated with cisplatin, ifosfamide, and paclitaxel (TIP) systemic chemotherapy for a planned total of four cycles. TIP treatment was complicated by a grade 3 peripheral neuropathy adverse event. CT imaging after the completion of treatment demonstrated a partial response of multiple lung nodules including a decrease of seven nodules and the complete resolution of one. After the planned four cycles, he was monitored with serial CT scans every 3 months. Radiographic progression occurred 13 months after TIP treatment was completed and revealed a new right liver lesion, the increasing size of multiple pulmonary nodules, and a new heterogenous pelvic enhancement involving the prostate and bladder, including the left ureterovesicular junction. In addition to radiographic progression, the patient reported increased pelvic pain and urinary frequency. At that time of progression, the patient was treated with ifosfamide and paclitaxel (IP). His persistent grade 1 neuropathy was felt by his physician team to be a contraindication for cisplatin use, particularly in combination with paclitaxel. He experienced stable disease for four months before developing progression of the pulmonary nodules and pelvic disease.

The patient remained interested in treatment and had a good functional status, with an Eastern Cooperative Oncology Group (ECOG) performance status of 1. Next generation sequencing was completed on the original penectomy specimen and showed no FDA-approved actionable alterations. The following pathogenic alterations were found: FBXW7 S476I, JUN amplification, MAPK1 E322K, MYCL1 amplification, and TSC2 E1583fs*14. The sample was microsatellite stable (MSS) and had a tumor mutation burden (TMB) of 0 Muts/Mb. The provider and patient discussed participation in a clinical trial, but ultimately the patient declined to enroll. Based on the limited standard of care regimens for advanced penile cancer, the off-label use of EV was identified as a potential therapeutic option. This was supported based on the mechanism of EV as an ADC against Nectin-4, which has high expression in squamous cell neoplasms [13,14], a documented response in urothelial carcinomas with squamous features [15], and staining on the penectomy specimen (outlined below).

The patient was treated with EV (1.25 mg/kg) on days 1, 8, and 15 of a 28-day cycle. The patient experienced marked improvement of his pelvic pain within one week of EV initiation, and the complete resolution of pain after completion of cycle 1. Of note, the patient had no other changes in his pain regimen or other analgesic measures such as palliative radiation. During cycle 2, the patient experienced worsening of his peripheral neuropathy from grade 1 to grade 2. Due to this adverse event, he was placed on a 3-week treatment break. The break improved his neuropathy, but delayed cycle 3. The patient also experienced a grade 1 rash, which responded to topical corticosteroids. A CT scan prior to the initiation of cycle 3 showed an interval decrease in the hepatic lesion, the resolution of the index pulmonary nodule, and improvement of the pelvic lesion (Figure 1).

Nectin-4 antibody (AB) staining was performed on the penectomy pathology specimen. Slides underwent IHC staining using the Leica Bond-RX and a commercially available anti-Nectin-4 AB (Cat# PA5-50463, ThermoFisher, Carlsbad, CA, USA). Nectin-4 expression was assessed by H-score based on staining intensity (0: none, 1: low, 2: medium, 3: high) and prevalence (0: 0%, 1: 1–33%, 2: 34–66%, 3: 67–100%). The IHC staining showed high Nectin-4 expression: IHC score was intensity (3+) and prevalence (3+) (Figure 2). IHC scoring and slide review was performed by a board-certified pathologist. This work was performed under VUMC IRB 191300.

The patient had restaging imaging after cycle 4, which showed a mixed response with progression of the lesion in the liver and stable disease in the lung and pelvic lesions. After a discussion of the risks and benefits, and given his ongoing pain control in the pelvis, the decision was made to continue treatment with EV and reassess the response with short interval monitoring. The repeat imaging after cycle 5 of EV showed further progression. The patient was transitioned to pembrolizumab but progressed after three cycles (2 months). Pembrolizumab was selected given its prior use in TMB-high penile cancer [16] and the evidence for its efficacy in other TMB-low tumors [17,18], including SCC [19], though a low likelihood of a response was acknowledged with this regimen. The patient then received gemcitabine [20,21] for two cycles (the patient declined any additional platinum chemotherapy) but the therapy was not tolerated. He then received panitumumab [22] for one cycle but his ECOG performance status declined quickly and he was transitioned to hospice.

## 3. Discussion

Penile cancer is an aggressive malignancy with poor prognosis and very limited effective treatment options for advanced disease. There is a critical need for novel therapeutic approaches in this cancer. However, the low incidence of cases limits the ability to study treatments in randomized clinical trials. In our case study, we have demonstrated the high Nectin-4 expression of a penile cancer tissue sample paired with the patient’s response as preliminary evidence to explore this class of therapy as a novel treatment for penile cancer.

Per NCCN guidelines, the preferred first-line therapy for metastatic penile SCC is paclitaxel/ifosfamide/cisplatin (TIP) [5], as cisplatin-containing regimens have shown the most activity in this disease [23]. 5-fluorouracil/cisplatin can be considered as an alternative first-line chemotherapy [24]. After progression on first-line chemotherapy, patients have overall poor responses. In a 30-patient cohort study of patients with stage TxN2-3M0 penile squamous cell carcinoma treated with TIP prior to lymphadenectomy, 19 patients progressed or recurred [25]. Seventeen of these patients received subsequent systemic therapy, with a median survival from first treatment failure of 5.7 months. Pembrolizumab can be considered in patients with microsatellite instability-high (MSI-high) tumors, as it has been studied in a tumor-agnostic manner and shown to have efficacy [26], [27]. There are case reports of the use of pembrolizumab in penile cancer that support the consideration of this treatment for MSI-high tumors or patients with high PD-L1 expression [16,28]. This is unfortunately a limited population, only estimated to be 15% of patients with advanced penile cancer [29]. There is case report support for combining immunotherapy treatment with radiation [30]. Cetuximab is also identified as a potential therapy based on a small retrospective study, in which 24% of patients had a partial response, and the median time to progression was 2.8 months [31]. There is also evidence that the HER2 pathway may have therapeutic implications for penile SCC [8]. A phase II study of dacomitinib, a pan-HER2 TKI which also inhibits EGFR, was conducted in patients with locally advanced (*n* = 20) or metastatic (*n* = 8) penile SCC [32]. In this trial, the overall response rate (ORR) was 32% for all patients, with a median progression-free survival of 3.2 months for patients with metastatic disease. Dacomitinib was shown to be most efficacious for patients with mutations in the downstream effectors of HER receptors or those with TERT mutations. These results, as well as the indications for immunotherapy, support genomic sequencing for patients at the time of progression, as important therapeutic targets may be identified.

Given the lack of robust standard treatment options, NCCN naturally recommends consideration for clinical trials. There are a number of ongoing trials for advanced penile cancer currently exploring classes of therapy including chemotherapy, VEGF/EGFR inhibitors, immunotherapies, and PARP inhibitors [6]. There are several ongoing trials exploring EV in other cancer types at present, some of which include squamous cell histologic subtypes. However, there are no studies evaluating a penile cancer cohort [33].

Nectin-4 is a type I immunoglobulin-like membrane protein found primarily in adherens junctions. It has been shown to affect cell proliferation [34], as well as tumor progression and metastasis [35]. The Nectin-4-directed ADC EV was first granted accelerated FDA approval in December 2019 and is approved for use in advanced or metastatic UC after progression on a PD-1 inhibitor and platinum-based chemotherapy [36]. The human IgG1 ADC binds to Nectin-4 on the cell surface and is internalized, allowing for the proteolytic cleavage of the microtubule-disrupting agent monomethyl auristatin E (MMAE) [12,37,38]. In a phase II trial of EV in UC, the ORR exceeded 50%; this ORR was particularly impressive given the significant pre-treatment with both platinum chemotherapies and PD-1/PD-L1 inhibitors [15]. Of note, in this trial, patients with UC with squamous features on histology were eligible and comprised 13% of the study population. Since its initial approval as monotherapy, EV has also been approved in combination with pembrolizumab for patients with locally advanced or metastatic UC who are ineligible for cisplatin-containing chemotherapy, and demonstrates a significant OS and PFS benefit in previously untreated locally advanced or mUC (including both cisplatin-eligible and -ineligible patients) [39].

EV use in UC does not require Nectin-4 positivity on tissue staining to qualify for treatment. However, numerous basic histopathology studies suggest that this therapy has great potential in other squamous neoplasms. Nectin-4 is highly expressed in squamous cell neoplasms including head and neck [40], lung [41], extramammary Paget’s [42], the squamous variant of muscle invasive bladder [43], and penile [9]. A recent study of primary penile cancers by Grass et al. demonstrated high Nectin-4 gene expression in the majority of penile cancer specimens: 70% of the samples had an at least 2-fold higher gene expression of Nectin-4 compared to normal glans [9]. This study also found 61% of the cases had moderate to strong Nectin-4 IHC staining. Importantly, recent preclinical data suggests that EV may have a “bystander” effect, targeting Nectin-4-negative cells in a mixed cellular assay, suggesting a potential role for this drug even in the setting of heterogeneous Nectin-4 expression [44]. There are also now numerous other drugs early in development targeting Nectin-4 including CAR-T therapy and novel ADCs [45].

In our patient, the response time to EV of 5 months was on par with other palliative systemic therapies, despite this patient’s substantial prior treatments. He tolerated the therapy well, with neuropathy and rash as the only reported side effects. EV treatment also provided substantial symptomatic improvement for him due to the response of the large pelvic lesion. Overall, the duration of the response and symptomatic improvement with low side effects provide support for EV use as a reasonable therapeutic option for metastatic penile cancer patients that are interested in systemic therapy if clinical trials are not an option. Of course, further studies should be performed to confirm these initial findings.

## 4. Conclusions

This case report provides promising evidence for the use of EV as a systemic therapy for metastatic penile cancer and reinforces prior work [9] demonstrating Nectin-4 expression in this rare cancer. To our knowledge, this is the first reported case of an ADC with a treatment benefit for penile cancer, including a symptomatic improvement and radiographic response. These findings may be important given the paucity of existing treatment options for this disease, hence, this warrants intensified basic research as well as the realization of clinical therapeutic trials.

## Figures and Tables

**Figure 1 ijms-24-16109-f001:**
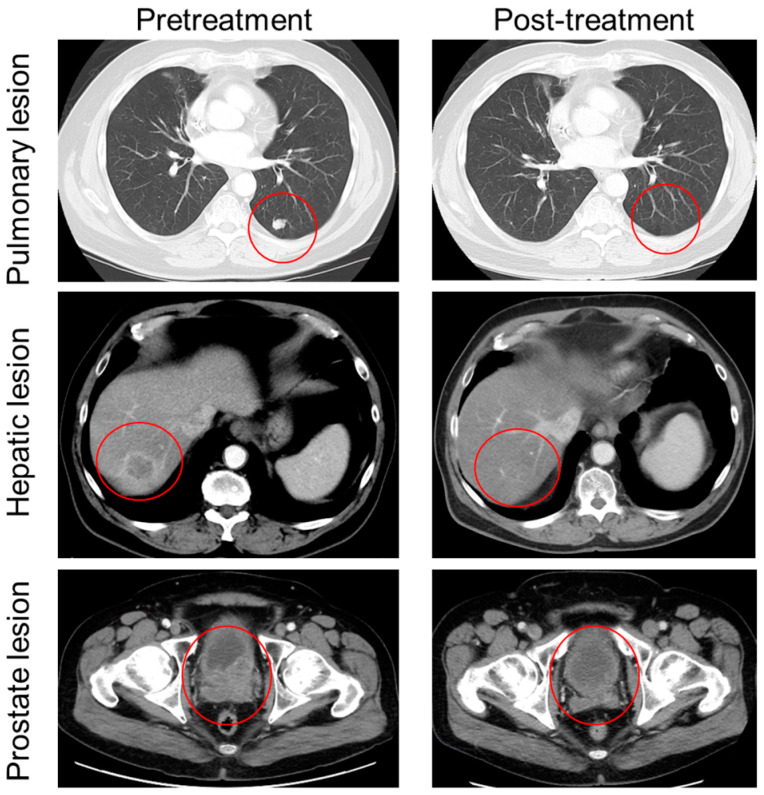
Radiographic response of patient with advanced penile carcinoma before and after treatment with EV. Pre-treatment images (**left**) taken 21 days before EV initiation. Post-treatment images (**right**) were obtained cycle 2 day 27. Resolution of index pulmonary lesion (**top**), near-resolution of R liver lesion (**middle**), and improvement in pelvic metastasis and concurrent posterolateral bladder wall invasion (**bottom**) are indicated by red circles.

**Figure 2 ijms-24-16109-f002:**
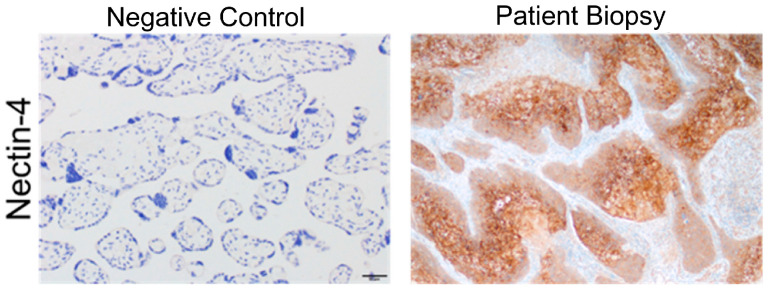
Nectin-4 expression in patient penile tumor. Immunohistochemistry staining of Nectin-4 in negative control (placental tissue, **left panel**) and patient specimen of penile tumor collected during penectomy (**right panel**). Magnification is shown at 20×; bar in left panel represents 50 µm.

## Data Availability

No new data were created or analyzed in this study. Data sharing is not applicable to this article.

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
