# Peer review of "Metastatic Penile Squamous Cell Carcinoma Responsive to Enfortumab Vedotin"

_ijms, 2023, doi:10.3390/ijms242216109_

Round 1

Reviewer 1 Report

Comments and Suggestions for Authors

In this manuscript, the authors present a case of metastatic penile squamous cell carcinoma with response to the Nectin-4 inhibitor EV. The patient had both radiographic and symptomatic improvement after 2 cycles of treatment, the patient was treated with EV (1.25mg/kg) on days 1, 8, and 15 of a 28-day cycle. A CT scan prior to initiation of cycle 3 showed interval decrease in the hepatic lesion, resolution of the index pulmonary nodule, and improvement of the pelvic lesion. This case report provides promising evidence for the use of EV as a systemic therapy for metastatic penile cancer. In general, this manuscript just presents a case report without novelty findings. So I think this manuscript is not suitable for publication in IJMS.

Specific comments:

1.    Figure 2, whether the authors can test the Nectin-4 expression after different cycle drug treatment.

Author Response

We appreciate the reviewer's thoughtful consideration of the merits of this case, and understand that as a case report, there is limited opportunity for scientific advancement of the field. We disagree that the findings lack novelty, as this is the first case examination of the use of EV in this cancer without alternative treatment options. Our hope is that the evidence presented in this manuscript can lead to future scientific advancement and provide the basis for more study of nectin-4 as a therapeutic target in other rare malignancies. 

Regarding figure 2, we unfortunately do not have access to serial biopsies for this patient, pre and post EV, as no additional biopsies were obtained. We agree it would be an interesting scientific question to determine if the nectin-4 expression decreases from time of response to time of progression, and we would support further evaluation of this question in a case series or prospective trial. The undertaking of such work would be supported by the publication of our manuscript. 

Reviewer 2 Report

Comments and Suggestions for Authors

This study was reported the case of metastatic penile squamous cell carcinoma responsive to enfortumab vedotin. The reviewer agrees to some of the content. However, the reviewer would like to suggest some opinions to make this paper as follows.

1.    Was TIP treatment discontinued due to adverse events?

2. Was the second-line therapy considered at the time of discontinuation of TIP treatment?

3.  The reviewer recognize that peripheral neuropathy is more likely with paclitaxel than cisplatin. Why was cisplatin omitted for second-line treatment?

4.   The reviewer would like to know the reason that the patients with microsatellite stable and tumor mutation burden of 0 Mets/Mb were received pembrolizumab.

Author Response

We thank the reviewer for their review and comments. We have addressed their individual questions below. 

  1.     TIP treatment was planned for a total of 4 cycles with surveillance after completion. This has been clarified in the updated manuscript

2.  There was no second-line treatment planned given TIP was stopped after a planned course of 4 cycles with plans for surveillance following. This has been clarified in the manuscript

3.   We agree with the reviewer that both paclitaxel and cisplatin can cause neuropathy, the primary physician felt that his neuropathy was prohibitive for cisplatin treatment, particularly in combination with paclitaxel. The manuscript has been updated to address this concern. 

4.   Pembrolizumab was selected given as a fourth line therapeutic, with the acknowledgement that response was less likely given his MSS-Stable and low TMB. While high TMB is predictive of response to ICI, low TMB patients can respond as well, with one single center analysis of 151 patients treated with ICI showing a response rate of 20% in TMB low patients. (Goodman et al, Molecular Cancer Therapeutics 2017). This citation and others showing response in low TMB populations have been added to the text. Rationale for therapy choice has been added to the text. 

Round 2

Reviewer 1 Report

Comments and Suggestions for Authors

the authors did not address my concerns, I suggest rejection

Author Response

The reviewer's comments are noted. We attempted to address the concerns in our prior response, and regret that the reviewer did not find this response sufficient.